# In Vitro, In Vivo and In Silico Assessment of the Antimicrobial and Immunomodulatory Effects of a Water Buffalo Cathelicidin (WBCATH) in Experimental Pulmonary Tuberculosis

**DOI:** 10.3390/antibiotics12010075

**Published:** 2022-12-31

**Authors:** Jacqueline Barrios Palacios, Jorge Barrios-Payán, Dulce Mata-Espinosa, Jacqueline V. Lara-Espinosa, Juan Carlos León-Contreras, Gerald H. Lushington, Tonatiuh Melgarejo, Rogelio Hernández-Pando

**Affiliations:** 1Instituto Nacional de Ciencias Médicas y Nutrición Salvador Zubirán, Departamento de Patología Experimental, Vasco de Quiroga 15, Belisario Domínguez Sección 16, Tlalpan, Ciudad de México 14080, Mexico; 2Qnapsyn Biosciences, Inc., Lawrence, KS 66044, USA; 3College of Veterinary Medicine, Western University of Health Sciences, Pomona, CA 91766, USA

**Keywords:** tuberculosis, cathelicidin, water buffalo, antimicrobial, immunomodulatory

## Abstract

Tuberculosis (TB) is considered the oldest pandemic in human history. The emergence of multidrug-resistant (MDR) strains is currently considered a serious global health problem. As components of the innate immune response, antimicrobial peptides (AMPs) such as cathelicidins have been proposed to have efficacious antimicrobial activity against *Mycobacterium tuberculosis* (*Mtb*). In this work, we assessed a cathelicidin from water buffalo, *Bubalus bubalis*, (WBCATH), determining in vitro its antitubercular activity (MIC), cytotoxicity and the peptide effect on bacillary loads and cytokines production in infected alveolar macrophages. Our results showed that WBCATH has microbicidal activity against drug-sensitive and MDR *Mtb*, induces structural mycobacterial damage demonstrated by electron microscopy, improves *Mtb* killing and induces the production of protective cytokines by murine macrophages. Furthermore, in vivo WBCATH showed decreased bacterial loads in a model of progressive pulmonary TB in BALB/c mice infected with drug-sensitive or MDR mycobacteria. In addition, a synergistic therapeutic effect was observed when first-line antibiotics were administered with WBCATH. These results were supported by computational modeling of the potential effects of WBCATH on the cellular membrane of *Mtb.* Thus, this water buffalo-derived cathelicidin could be a promising adjuvant therapy for current anti-TB drugs by enhancing a protective immune response and potentially reducing antibiotic treatment duration.

## 1. Introduction

Tuberculosis (TB) has been a prevalent infectious disease since ancient times and remains a worldwide major health problem [1]. With 10 million active cases and 1.4 million deaths annually, TB morbidity represents the most significant number of incidences and human deaths attributable to a single bacterial agent [2]. Its causal agent, *Mycobacterium tuberculosis* (*Mtb*), can survive in a latent state in infected individuals, thereby serving as a reservoir, awaiting the reactivation that usually occurs in immune-suppressed individuals, such as older people, HIV, and diabetic patients [1]. According to the World Health Organization estimation, 2 billion people, almost one-quarter of the world’s population, are latent infected [1]. From this vast population, it is estimated that 10% will reactivate progressive suffering disease. Another significant factor is the continued emergence of new multidrug-resistant strains (MDR), often associated with poor compliance due to the long, complex, toxic and expensive treatment [3]. The therapeutic regimens for MDR-TB, in vitro resistance to at least isoniazid and rifampicin, should be prescribed for at least 18 months [3]. Thus, new anti-TB drugs and better therapeutic strategies are urgently needed. New therapeutic candidates should short the standard regimens and ideally be effective against MDR strains. Hence, active research aiming at developing new TB drugs has intensified in the last few years [4,5].

The immune system is a critical factor in preventing mycobacterial infection. During primary *Mtb* infection, bacilli growth is usually well controlled by the immune system. Some infected persons, perhaps not more than 5 to 10 per cent, develop active progressive disease [4]. It is not entirely known why some people develop progressive TB whereas others do not; perhaps in sick individuals several factors of the innate or acquired immune system are inefficient or are not sufficiently expressed to fend against the *Mtb*. The respiratory tract is the main entrance of *Mtb.* Airway epithelia and leukocytes are important protagonists of the initial innate defense response, which is in part mediated by the production of cytokines and peptides with antimicrobial activity (AMP), such as defensins and cathelicidins. These immune mechanisms not only play a critical role during early infection but also, during late disease when are generally overproduced in trying to defend the infected individual [3,4].

The two main mammalian AMPs, defensins and cathelicidins, are small, cationic, amphipathic molecules ascribed initially with a pore-forming activity in prokaryotic cells. Further research has shown that these novel peptides possess chemotactic, wound-healing and immunomodulatory activities [6]. Therapeutic strategies based on AMP induction of bacterial lysis, chemotaxis, autophagy, or regulation of the immune response are well known [6,7], also their mechanism of action is independent of the type of bacterial strain. Thus, AMPs represent a potential immunotherapeutic strategy for *Mtb* infection [8].

Cathelicidins are an ancient class of AMPs with a broad spectrum of bactericidal activities, first identified in bovine neutrophils [9]. They are characterized by a highly conserved N-terminal signal peptide, a pro-region (cathelin domain), and a highly heterogeneous C-terminal antimicrobial domain [10]. Cathelicidins are initially translated as precursor pre-pro-peptides and are released as active AMPs after proteolytic cleavage. The mature peptides are rich in certain amino acids, such as proline, arginine, and tryptophan and exhibit a variety of secondary structures, i.e., α-helix, β-hairpin, and random coils [11]. These amphipathic peptides bind to the bacterial membrane and kill bacteria by different mechanisms, such as water-filled toroidal pores [12,13], disruption of cytoplasmic membrane or inhibition of DNA and protein synthesis [13]. Cathelicidins are a prominent component of the antimicrobial capability of neutrophils, macrophages, mast cells and epithelial cells [14,15]. They are also crucial elements in the inherent defense mechanisms within epithelial cells of the lung, urinary bladder, gut, oral mucosa, skin, and testis [16,17].

Cathelicidins have been isolated and characterized in many placental mammals, such as primates (human, rhesus monkey) [18,19], Artiodactyla (cows, sheep, goats, pigs) [20,21,22], Perissodactyla (horses) [23], Rodentia (mice, rats, guinea pigs) [24,25], Lagomorpha (rabbits) [26], and carnivores including dogs that are highly resistant to infection with diverse microorganisms. Thus, cathelicidins play an essential role in the innate immunity of diverse animals. Seemingly, animals living in harsh environments with a high load of environmental pathogens have evolved their immune systems towards a more efficient AMP production, protecting themselves and reducing the spread and pathogenicity of infectious diseases. It is apparently the case with water buffalo (*Bubalus bubalis*), considering that these animals live in the highly contaminated swampy areas of tropical and subtropical climates. Humans and mice only have one type of cathelicidin, while water buffalo has seven, because of gene duplication that probably reflects pathogen-driven natural adaptation to their harsh ecological niche [27]. Furthermore, compared to cattle, water buffalo is resistant to diverse infectious diseases, such as tick-borne diseases, trypanosomiasis, brucellosis, and TB [28,29]. The present work aimed to evaluate the effect of water buffalo cathelicidin (WBCATH) on *Mtb* in vitro, in vivo, and in silico using alveolar macrophages, a mouse model of progressive pulmonary TB, as well as computational modeling of the WBCATH.

## 2. Results

### 2.1. WBCATH Bactericide Activity against Mtb In Vitro

The WBCATH bactericidal activity was tested in vitro against the *Mtb* reference strain H37Rv and the MDR clinical isolate CIBIN-99. H37Rv is susceptible to all five first-line anti-TB drugs: streptomycin (STR), pyrazinamide (PZA), isoniazid (INH), rifampicin (RIF) and ethambutol (EMB). CIBIN-99 is an isolate from a patient with advanced pulmonary TB that is resistant to all the above-mentioned antibiotics and was previously fully characterized [30]. Both strains were incubated with different concentrations of WBCATH, and due to their high antibiotic efficiency, INH was selected as the positive control primary antibiotic and Amikacin (AMK) was the positive control of second-line antibiotics. In all the concentrations tested there was a significant microbicidal effect, from 1 to 16 μg, being maximal at 32 and 64 μg (Figure 1A,C). These results were well related to the number of colony-forming units (CFU) The CFU from the mycobacteria H37Rv incubated with WBCATH at 16 μg/mL, decreased by 79.18%, the 32 μg/mL 84.33% and the 64 μg/mL 85.60% compared to the untreated control (UTC), respectively. In the MDR strain, the treatment with 16 μg/mL of WBCATH decreased by 54.22% the CFU, 32 μg/mL by 71.16% and 64 μg/mL by 98% compared to the UTC. The treatment with 32 and 64 μg/mL was more effective in decreasing the CFU than the group treated with AMK (positive control), and the effect was dose-dependent. These results confirmed the efficient microbicidal effect of WBCATH against drug-sensible and MDR strains (Figure 1B–D and Figure 2B–D).

### 2.2. WBCATH Affect the Ultrastructural Morphology of Drug Sensible and MDR Mtb

WBCATH at 32 and 64 μg concentrations after 18 h of incubation with drug-sensible and MDR *Mtb* produced many structural abnormalities, such as multiple cytoplasmic vacuoles (Figure 2A), cell wall swollen with segmental disappearance (Figure 2B), spherical electron-dense bodies surrounded by electron-dense layer and a clear halo with irregular electron-dense material dispersedly distributed in the cytoplasm (Figure 2C), as well as disruption and fragmentation of the cell wall and peripheral membrane (Figure 2D). These results confirm an efficient lytic effect of WBCATH cathelicidin on *Mtb*.

### 2.3. WBCATH Do Not Affect Alveolar Macrophage Survival In Vitro and Induces Hemolysis in High Concentrations

WBCATH cathelicidin cytotoxicity was evaluated in alveolar macrophages, considering the in vitro MIC values. WBCATH cell toxicity was very low at 10 to 160 μg/mL concentrations, showing more than 90% macrophage survival (Figure 3A). Therefore, we concluded that 160 µg/mL of WBCATH has negligible toxicity effects against alveolar macrophages and this concentration was used in the following in vitro experiments.

To extend the WBCATH cytotoxicity study, the hemolytic activity of the WBCATH at different concentrations was screened against normal human erythrocytes. Hemolytic activity is expressed in % hemolysis and reported as the mean ± standard error of the mean of four replicates. WBCATH at a concentration of 10–20 µg/mL produced low hemolytic activity, higher hemolysis percentage was found with an increase in peptide concentration (Figure 3B).

### 2.4. WBCATH Decreased the Bacterial Burden in Macrophages Infected with Drug-Sensitive or Drug-Resistant Mtb In Vitro

The non-toxic WBCATH concentration of 160 µg/mL in non-infected alveolar macrophages was used to assess the effect of *Mtb* killing in infected alveolar macrophages. In comparison with the control non-treated infected macrophages, WBCATH decreased by 86.27% the bacillary load 24 h after infection, 87.02% after 72 h and 74.85% after 144 h (*p* < 0.0001). This result demonstrates that WBCATH improves bacterial killing by alveolar macrophages.

During infection, there is macrophage activation and cytokine production, tumor necrosis factor-alpha (TNFα) and interleukin 12 (IL-12) are significant protective cytokines. Thus, 1 × 10^5^ infected and non-infected macrophages plated in 12-well dishes were incubated with or without WBCATH (16 µg/100 µL) and supernatants were collected after one and 24 h to assess TNFα and IL-12 production determined by ELISA. WBCATH induced a subtle nonsignificant increase of TNFα after one and 24 h in non-infected macrophages, while a significant two-fold higher concentration of TNFα was determined after 24 of infection in macrophages incubated with WBCATH in comparison with the other groups (Figure 4). WBCATH induced three fold higher production of IL-12 after one and 24 h of incubation with infected and non-infected alveolar macrophages (Figure 4). Thus, WBCATH induces the production of TNFα and IL-12 in infected macrophages and IL-12 even in non-infected macrophages.

### 2.5. WBCATH Has a Therapeutic Activity in the Murine Model of Progressive Pulmonary TB with Drug-Sensitive or Drug-Resistant Strains

The mean MIC of WBCATH was 32 μg/mL, this concentration was administered intratracheally (i.t), three times per week in mice with late progressive TB produced by either strain. After 60 days post-infection, control mice treated only with the vehicle (saline solution, SS) showed high bacillary loads and tissue damage (pneumonia), which is consistent and characteristic of late and active TB (Figure 4). In contrast, i.t. administration of WBCATH in animals infected with drug sensible strain H37Rv significantly reduced lung bacillary loads at day 28 and 60 post-treatment, in comparison with control animals three and fivefold lesser, respectively (Figure 5A). A similar response was seen in mice infected with the MDR strain, WBCATH treatment induced fivefold lesser CFU than in control animals (Figure 5D). Infected animals with H37Rv showed a significant decrease in pneumonia (Figure 5B,C), while in treated mice infected with MDR there was more pneumonia than in the control group (Figure 5E,F), but in both treated groups with WBCATH was seen wide perivascular lymphocyte infiltrates, particularly around venules, resembling lymphoid follicles (Figure 5C–F).

### 2.6. WBCATH Induces Higher Expression of Protective Cytokines in Tuberculous Mice Infected with Drug Sensible or MDR Strains

To investigate the role of WBCATH as a regulator of protective immune responses in experimental pulmonary TB, cytokine gene expression was evaluated by RT-PCR and protein production by immunohistochemistry. In comparison with the control non-treated mice, after two months of WBCATH treatment, the gene expression of TNFα was significantly higher in mice infected with H37Rv, while the expression of IFNγ was significantly higher after one month of treatment (Figure 6). To study the production of these cytokines at the protein level, both cytokines were detected by immunohistochemistry and the percentage of immunostained cells was determined, particularly in the inflammatory infiltrate around blood vessels and airways, and in pneumonic areas. Treated animals showed a significantly higher percentage of lymphocytes positive to IFNγ around blood vessels and airways, as well as in the airway epithelium, while TNFα immunostained macrophages were significantly more numerous in areas of pneumonia (Figure 6). Similar results were seen in animals infected with MDR *Mtb* (Figure 7).

### 2.7. WBCATH Had a Synergistic Therapeutic Effect in Combination with First-Line Antibiotics

To determine the synergistic activity of WBCATH with first-line antibiotics (isoniazid, rifampicin, streptomycin), a group of mice after two months of infection with drug sensible strain H37Rv was treated with this chemotherapy scheme plus WBCATH and compared with mice treated only with antibiotics (AB). The group WBCATH + AB significantly decreased bacillary loads compared with the group that received only antibiotics, but both groups showed similar pneumonia (Figure 8). The decrease of bacilli burdens in the combined treatment group began one-week post-infection, suggesting that this combined treatment could shorten conventional chemotherapy.

### 2.8. Computational Modeling of the Potential Effects of WBCATH on the Cellular Membrane of the Mtb

Molecular dynamics simulations of the WBCATH dimer were performed using the protocols described in [U1] Section 4.14. These simulations suggest that starting from a position distinct from the membrane bilayer, the peptides should quickly associate with the surface and, within the period of simulation, could excavate a cavity with a depth of roughly 7 Å below the median plane of undulation of the bilayer surface (Figure 9B). While the simulation was not long enough to fully reproduce membrane penetration, structural dynamics did reveal several key incremental roles achieved by specific amino acids within the WBCATH structure. In particular, the W139 and F142 residues on both monomers collaborated dynamically as ‘excavator’ tools to pry apart membrane lipids to admit fuller lipophilic binding by peptide aliphatic (Leu and Ile) side chains to advance stable peptide integration into the layer. Electrostatically, the simulation further suggests that R143 may be crucial for establishing initial attractions with membrane electronegative phosphatidyl groups, while R137 has a dual role of also fostering cross-monomer interactions with the peptide carboxy terminus for oligomer stabilization.

Using the protocols described in [U2] Section 4.14 simulations for the WBCATH oligomeric pore candidates commenced with the peptide oligomers embedded into the bilayer in anti-parallel membrane-spanning orientations. Since, by definition, anti-parallel pores entail an even number of oligomers, and since stable pore sizes are generally determined by a compromise between strain (which precludes the smallest pores) and stability (which degrades in large flexible structures, the study focused on hexamer, octamer and decamer pores. During the simulations, the octameric peptide structure sustained an ordered arrangement (Figure 9C), reminiscent of a physiological pore throughout the full course of the simulation. The hexameric and decameric structures tended to collapse into disorder, however.

## 3. Discussion

Widespread and indiscriminate use of antibiotics, in humans and animal production, has heavily contributed to the emergence of antimicrobial-resistant microorganisms. In several types of bacteria, a highly antibiotic resistance have been observed with increasing incidence over the past several decades [31]. In the case of TB, the lack of compliance due to the long and multidrug treatment is the principal factor to induce the emergence of drug-resistant bacteria. Therefore, novel approaches are required to address the problem of bacterial antimicrobial resistance (e.g., antimicrobial peptides, and natural products).

Mammals have evolved multiple defense mechanisms to survive in an environment filled with pathogens [32]. AMPs play an essential role in the defensive mechanisms of the innate immune system against bacterial, fungal, and viral infections [33]. Although AMPs were initially described as broad-spectrum antimicrobials, they have many other discovered functions, such as chemotaxis, activation and maturation of phagocytic cells, promotion of proinflammatory responses, apoptosis regulation, angiogenesis, and wound healing, among others [34]. Cathelicidins are cationic peptides found in leukocytes and epithelial cells that play a central role in the innate immune response against bacteria, viruses and fungi, particularly airborne pathogens [27,35,36].

Cathelicidins are a vital AMP that has been suggested to be involved in the control of TB mainly through its direct antimicrobial activity. Some studies have demonstrated the contribution of cathelicidins in the elimination of *Mtb*, showing that the main cells that produce it during infection are alveolar macrophages [37,38]. On other hand, it is plausible that more efficient bactericidal cathelicidins should be produced by animals that live in harsh environments, such as water buffalos that live submerged in swamps filled with loads of potential microbial pathogens. Accordingly, the present study assessed, in vitro in vivo, and in silico, the effect of one synthetic cathelicidin from water buffalo against *Mtb*. Supporting this idea is the fact that water buffalo (*Bubalus bubalis*) is an important livestock for milk and meat production that has improved resistance against many infectious diseases that affect beef and dairy cattle [34]. This innate resistance may be explained in part because they express numerous AMPs, such as defensins, cathelicidins, and hepcidin, which play an essential role in the protection against several important pathogens [34].

Our results showed a dose-dependent direct effect of WBCATH on the viability of *Mtb* in vitro using the reference strain H37Rv and in an MDR clinical isolate. This effect was observed using doses from 1 to 64 µg/mL that were not toxic for alveolar macrophages. The electron microscopy studies confirmed that WBCHAT has a lytic effect on *Mtb*. However, this bactericidal effect could be variable depending on the infecting strain, considering the high diversity of the *Mtb* genotypes that should be related to significant differences in the biochemical bacterial constitution and functional activities. We observed that in vitro the MDR isolate was more susceptible that the reference strain H37Rv.

Different cathelicidins from *Bubalus bubalis* also showed antimicrobial activity against *E. coli*, *S. aureus* and *C. albicans* [39]. Numerous factors define the antimicrobial properties of AMP, including their cationic charge which is possibly the most important of these factors. Since the surface of the bacterial membrane is negatively charged, AMP with a more positive charge shows higher affinities to the bacterial membranes, disruption of the membrane is a common mechanism of action of the AMP [39], as our electron microscopy study showed. Therefore, the decreased viability of *Mtb* caused by WBCATH could be related to direct damage induced in the bacteria membrane by this high cationic cathelicidin.

Following the confirmation that WBCATH is not cytotoxic at concentrations lower than 160 μg/mL in murine macrophages, we observed that treatment with WBCATH at a concentration of 32 or 16 μg/mL induced an essential reduction of CFU in infected macrophages with the drug-sensitive strain. Thus, these results suggest that low doses of WBCATH in vitro could enhance the protective immune response against mycobacterial infection mediated by macrophages. Similar results have found that exogenous addition of LL-37, or endogenous over-expression of cathelicidin in macrophages, significantly reduced the intracellular survival of mycobacteria (*M. smegmatis* and *M. bovis* BCG infections). This was done by a direct effect on the mycobacteria or by inducing phagolysosomal formation [40]. Our in vivo studies showed that therapy with WBCATH decreases pulmonary bacillary loads and pneumonia while inducing a high proinflammatory cytokine expression. Thus, these significant protective effects could result from WBCATH peptide direct antimicrobial activity and the induction of a protective Th1 immune response. Indeed, we observed that TNFα and IL-12, which are significant cytokines to induced Th-1 response, are highly induced by WBCATH in mycobacterial infected macrophages in vitro, and IL-12 is even highly induced by WBCATH in non-infected alveolar macrophages. Regarding the other significant protective cytokine IFNγ we observed that its expression is different in both strains when they are treated with WBCATH. In Figure 6B, INFγ increases significantly at 1 month and subsequently decreases, however, in the MDR strain (Figure 7B), there is a decrease during treatment. This could be because the immune response that each strain causes is different. IFN-γ is a very important cytokine in the control of infections caused by intracellular bacteria such as *Mtb*. In human pulmonary Tb, it has been described that there is a relationship between the production of IFNγ and the clinical manifestations of the disease; the more severe the disease, the lower levels of IFNγ are produced by peripheral blood mononuclear cells. Additionally, when WBCATH was administered with first-line chemotherapy, a quicker decrease of bacillary load and tissue damage was seen than in mice treated only with antibiotics. The majority of AMP interact with the bacterial membrane producing certain damage and thereby could facilitate antibiotics’ entrance into the cell and interaction with their intracellular targets [41], as described for Nisin Z, and Ocellatin-PT3 [42,43]. This suggests that WBCATH can shorten conventional chemotherapy. This is important to TB control since one of the primary disadvantages of the current chemotherapy is its long duration, resulting in poor compliance, recidivism, toxicity, and the development of drug-resistant strains.

Other studies showed that LL-37 (human cathelicidin) also inhibits the IFNγ priming of lipopolysaccharide (LPS) responses and the synergistic response to combined treatment with IFNγ and LPS, suggesting an important modulatory function of this peptide not only promoting inflammation but balancing the response to avoid exacerbated inflammation [44]. These suppressive effects of LL-37 on IFNγ responses were mediated through the inhibition of STAT1-independent signaling events, involving both the p65 subunit of NF-κB and p38 mitogen-activated protein kinase (MAPK) [45]. Cathelicidin can also attract lymphocytes and potentiate their activation [46], our results suggest that this was a significant WBCATH activity considering the higher expression of IFNγ and TNFα expressed by numerous lymphocytes around blood vessels and in macrophages of pneumonic areas, respectively.

Simulations of the octameric antiparallel WBCATH oligomer are strongly suggestive of the capacity of the WBCATH peptide to oligomerize in a manner that spans a lipid bilayer model of a microbial cell membrane. This may be a plausible explanation for experimental observations of WBCATH antimicrobial efficacy, in that the pores could alter membrane homeostasis. Indeed, short, helical peptides with the right balance of amphiphilicity (i.e., mostly hydrophobic, but with a contiguous region of net cationicity) are well suited to prospective broad-spectrum interference with bacterial membranes, whose lipid content tends to favor long-chain alkyls with ensemble net negative charges. This collectively presents a possible adjuvant antibiotic scheme that is independent of conventional proteomic targets for which many bacteria rapidly evolve resistance. It is nonetheless notable, though, that emerging research has observed a tendency for bacteria to modulate membrane electrostatics in a manner that quite possibly relates to surviving harmful electrostatic threats, suggesting that microbial vulnerability to membrane targeting may also be affected by evolutionary resistance [47].

In summary, the use of WBCATH administered alone and in combination with anti-TB antibiotics was analyzed under different conditions, including computational modeling. Taken all together, our results showed that WBCATH reduced mycobacterium viability and induced the expression of the pro-inflammatory cytokine TNFα, related to Th1 immune response, which significantly contributed to controlling infections with drug-sensitive and drug-resistant *Mtb*. In addition, WBCATH potentiated the effect of first-line antibiotics. Thus, although this is an animal model experiment and only two *Mtb* strains were tested that showed different pro-inflammatory effects induced by WBCATH, our results suggest that WBCATH could be a potential complementary treatment for this devastating infectious disease.

## 4. Materials and Methods

### 4.1. Reagents

The Middlebrook 7H9 and 7H10 media and the OADC (oleic acid, albumin, dextrose and catalase) were obtained from Becton-Dickinson (Detroit, MI, USA). Cell Titer 96^®^ Aqueous One Solution Cell Proliferation assay reagent was obtained from Promega (Madison, WI, USA). The NucleoSpintotal RNA FFPE kit for RNA extraction was obtained from Macherey-Nagel Thermo Fisher Scientific (Waltham, MA, USA). The Omniscript^®^ Reverse Transcription Kit for obtaining complementary DNA and the QuantiTectTM SYBR^®^ for RT-PCR were obtained from Qiagen (Germantown, MD, USA). The primers of the analyzed cytokines were obtained from InvitrogenTM Thermo Fisher Scientific (Waltham, MA, USA). Polyclonal rabbit antibodies anti-IFNγ and TNFα were purchased from Santa Cruz Biotechnology (Santa Cruz, CA, USA). All other reagents were of analytical grade and were obtained from known commercial sources.

### 4.2. Experimental Design

We evaluated the effect of WBCATH on *Mtb* growth in vitro. First, we established the Minimum Inhibitory Concentration (MIC) against *Mtb* drug sensible (H37Rv) ATTC 27294, susceptible to all five first-line anti-TB drugs (streptomycin, INH, rifampin, ethambutol, and pyrazinamide), and a clinical isolate that is resistant to all of these drugs (Table 1), CIBIN 99 (MDR), which was isolated, identified and characterized in the Mycobacteriology Laboratory of the Centro de Investigación Biomédica del Noreste (CIBIN), Instituto Mexicano del Seguro Social (IMSS) in Monterrey, Nuevo León, Mexico [30]. and the changes in the bacteria morphology by electron microscopy. Then, we evaluated the effect of WBCATH on the cell viability of murine macrophages, and their effect on the clearance capacity and production of cytokines in non-infected and macrophages infected with either H37Rv or MDR strains. In the in vivo experiments, we used a well-characterized murine model of progressive pulmonary TB [48]. Animals were infected with H37Rv or MDR, treated on day 60 post-infection with WBCATH and euthanized after one- and two months post-treatment; lungs were used to determine bacillary loads by colony forming units (CFU), the extension of tissue damage (pneumonia) and cytokine expression by immunohistochemistry, determining by automated morphometry the percentage of positive cells for each cytokine in specific lesions (pneumonia, perivascular inflammatory infiltrate). A similar experiment was done with first-line antibiotics (Figure 10). Animals were monitored daily and were humanely euthanized under pentobarbital anesthesia if respiratory insufficiency, accentuated cachexia, or total immobilization was noted.

### 4.3. Minimum Inhibitory Concentration (MIC) Determination

An initial load of 1.5 × 105 bacteria was inoculated per well in Middlebrook 7H9 broth (Difco Laboratories, Detroit, MI, USA) at a final volume of 200 μL. The cultures were incubated with serial dilutions of WBCATH, ranging from 1 to 64 μg/mL. Wells containing only broth were used as sterility control, and wells containing only bacteria were used as growth control. Isoniazid or amikacin was a negative growth control (inhibit bacterial growth). Plates were incubated at 37 °C with 5% CO2 for seven days in a humidified incubator. Changes in bacillary growth were measured with the Cell Titer 96^®^ Aqueous One Solution Cell Proliferation assay reagent (Promega, Madison, WI, USA) which is a colourimetric method. Based on measuring cell viability, the substrate used is 3-(4,dimethylthiazol-2-yl)-5-(3-carboxymethoxyphenyl)-2-(4-sulfophenyl)-2H-tetrazolium, or MTS, if the bacterium is viable, can reduce the substrate through the active dehydrogenase enzymes that it possesses and generate the formazan product, which can be measured in a spectrophotometer at an optical density of 490 nm and is proportional to the number of live bacteria. After five days of incubation at 37 °C, the MIC was determined as the lowest concentration of the agent that completely inhibits visible growth. An MTS mixture (20 μL/well) was added four hours before each exposure period. After completing the exposure period, MIC values were determined spectrophotometrically at 570 nm (BioTek Instruments, ELX 800, Winooski, VT, USA). The antimicrobial activity was confirmed from broth dilution MIC tests by subculturing agar plates that do not contain the test agent. Colony-forming units (CFU) were determined to verify the result after 21 days of culturing.

### 4.4. Electron Microscopy Study

Transmission electron microscopy was used to study the ultrastructural damage to *M. tuberculosis* caused by treatment with WBCATH. Briefly, bacilli (*Mtb*, H37Rv) were cultured in Middlebrook 7H9 broth (Difco Laboratories) supplemented with Middlebrook OADC enrichment media (BBL; BD, Franklin Lakes, NJ, USA) until the logarithmic phase was achieved. Then, viable bacilli (1 × 10^7^) were placed in the wells of 96-well plates and were exposed to WBCATH for 18 h at the MICs determined previously. Subsequently, fixation was performed with 1% glutaraldehyde dissolved in 0.1 M cacodylate buffer (pH 7); postfixed in 2% osmium tetraoxide for 4 h, and the fixed bacilli suspension was washed three times with cacodylate buffer, dehydrated with increasing concentrations of ethanol and gradually infiltrated with Epon resin (Pelco, Hawthorne, CA, USA). Thin sections were contrasted with uranyl acetate and lead citrate (Electron Microscopy Sciences, Fort Washington, PA, USA), and examined with an FEI Tecnai transmission electron microscope (Hillsboro, OR, USA).

### 4.5. Cytotoxicity Activity Tested In Vitro

Cytotoxicity of both cathelicidins was evaluated by crystal violet staining [49]. Briefly, murine alveolar macrophages (cell line MHS) were plated in 96 multiwell tissue culture plates and cultured for 48 h with different K9CATH and WBCATH concentrations (1–64 µg/mL). Then, cells were fixed with 1.1% glutaraldehyde for 10 min. Plates were washed with deionised water, dried, and stained for 10 min with 0.1% violet crystal. Next, cells were solubilised with 200 µL of 10% acetic acid, incubated in agitation for 30 min and read in a plate spectrophotometer at 590 nm.

### 4.6. Hemolysis Assays In Vitro

A hemolytic assay was carried out in freshly collected human red blood cells. Briefly, blood was taken and washed three times with 0.9% NaCl (3000 rpm for 3 min). The plasma was removed and the cells were suspended in 0.9% NaCl at a final concentration of 30% of the total hematocrit. 100 μL of the suspension cells were plated in a 96-well plate. Seven different concentrations of WBCATH were used (10–640 μL WBCATH) and were mixed with 100 μL of 0.9% NaCl. The reaction mixture was placed in an incubator for two hours at 37 °C. After the incubation time, the reaction mixture was centrifuged at 3000 rpm for 3 min. The supernatants (containing haemoglobin released from lysed erythrocytes) were measured at 550 nm. Deionized water was used as a positive control and 0.9% NaCl as the negative control (UTC). The experiment was done in quadrupled and the percentage of hemolysis was calculated in comparison with the UTC. Data are presented as mean ± SEM of the four replicates.

### 4.7. Evaluation of Antimycobacterial Intracellular Activity and Cytokines Production by Alveolar Macrophages

The antimycobacterial intracellular activity was tested in the murine macrophage cell line MHS infected with *Mtb* H37Rv, using non-toxic concentrations of cathelicidin (WBCATH 160 μg/mL). Log-phase growth of *M. tuberculosis* H37Rv in Middlebrook 7H9 broth with 10% OADC and 0.05% of tyloxapol were washed twice with saline solution and adjusted in RPMI to reach a bacterial macrophage multiplicity of infection of 1:5. One day before the infection, 1 × 10^4^ macrophages per well were plated in 12 multiwell tissue culture plates, incubated for one hour with WBCATH and exposed to mycobacteria. Non-phagocytosed bacilli were removed by three washes with warm RPMI plus antibiotics (Penicillin-Streptomycin Thermo Fischer Scientific, Waltham, MA, USA). Then, at 1, 24, 36 and 76 h, the cells were lysed with 100 µL of 0.1% sodium dodecyl sulfate (SDS) for 10 min and later,100 μL of 20% bovine serum albumin (BSA) was added. Control cells contained only the culture medium. Viable bacteria were determined by quantifying CFU by plating dilutions of the macrophage lysates in Middlebrook 7H10 agar. The colonies were counted at 14 and 21 days.

Supernatants from non-infected and infected macrophages incubated with WBCATH were obtained after one and 24 h. of infection and used to determine TNF and IL-12 by ELISA the TNF and IL-12 Quantikine ELISA Kit (BD Pharmingen, San Diego, CA, USA, No. CAT 558534, No. CAT 555165, respectively), according to the manufacturer’s instructions.

### 4.8. Experimental Model of Progressive Pulmonary Tuberculosis in BALB/c Mice

The experimental model of progressive pulmonary TB has been described in detail before [50]. Briefly, pathogen-free male BALB/c mice, 6–8 weeks of age, were anesthetized (Sevoflurane; Abbott Laboratories, Chicago, IL, USA) and infected with a high dose to induce progressive disease with strain H37Rv or CIBIN99 by endotracheal route (i.t.), administering 2.5 × 10^5^ viable bacteria suspended in 100 μL of PBS. Infected mice were maintained in groups of five in cages fitted with microisolators connected to negative pressure. All procedures were performed in a biological security cabinet at a Biosafety level III facility. All the animal work was carried out according to the guidelines and approval of the Ethical Committee for Experimentation in Animals of the National Institute of Medical Sciences and Nutrition (INCMNSZ) in Mexico City, permit number CINVA 1825 PAT-1825-16/18-1.

### 4.9. WBCATH Administration Alone or with First-Line Antibiotics

To evaluate the effect of WBCATH in *Mtb* in vivo, animals surviving 60 days after infection with drug-sensitive strain H37Rv or MDR strain were randomly allocated into two treatment groups: (1) animals treated by intratracheal route (i.t.) every other day with 32 μg of cathelicidin dissolved in 50 μL of saline solution (SS) as the vehicle. (2) infected mice that only received the SS as a control group under the same procedure. Another experiment to evaluate synergy with antibiotics included three treatment groups: (1) animals treated i.t. every other day with 32 μg of WBCATH dissolved in 50 μL of saline solution plus the first-line antibiotics 10 mg/kg rifampicin (RIF), 10 mg/kg isoniazid (INH), and 30 mg/kg pyrazinamide dissolved in SS. (2) animals treated daily with the same first-line antibiotics scheme. (3) infected mice that only received SS as a control group. Groups of six animals were euthanized by exsanguination under anesthesia with intraperitoneal (i.p.) pentobarbital, after 30 and 60 days post-treatment, the lungs were immediately removed; the right lung was frozen by immersion in liquid nitrogen and used for CFU determination, while the left lung was perfused for histopathology analysis. Two independent experiments were performed.

### 4.10. Determination of Pulmonary Bacillary Loads by Colony-Forming Units (CFU)

For CFU determination, frozen lungs were disrupted using ceramic beads in tubes with 1 mL of PBS containing 0.05% tween 20. Four dilutions of each homogenate were spread onto duplicate plates containing Bacto Middlebrook 7H10 agar (Difco BD, Sparks, MD, USA) enriched with OADC (Difco, Sparks, MD, USA). The incubation time was 21 days.

### 4.11. Preparation of Tissue for Histology and Morphometry

For the histological study, parasagittal sections were taken through the hilum, dehydrated, and embedded in paraffin, sectioned at 5-μ width, and stained with Hematoxylin and Eosin (H&E). The percentage of lung area affected by pneumonia was measured using a Leica Q-win Image Analysis System (Cambridge, UK).

### 4.12. Cytokine Expression by Immunohistochemistry

The same paraffin-embedded tissues were used for immunohistochemistry. Sections 5 μm width were mounted on slides covered with poly L-lysine (Biocare Medical, Lake Concord, CA, USA). For dewaxing, the slides were placed at 60–70 °C for 20 min and then incubated for 5 min into xylene. The slides were changed five times into the medium in the following sequence: (i) xylene-alcohol (1:1), (ii) absolute alcohol, (iii) alcohol 96%, and (iv) distilled water. Once hydrated, endogenous peroxidase was blocked with methanol–10% H_2_O_2_. The washings were performed with HEPES-buffered saline (HBS)-Tween 20 (10 mM HEPES, 150 mM NaCl, 2 mM CaCl_2_, 0.5% Tween 20). The tissue areas were delineated and then blocked with 100 μL of HBS with 2% background sniper (BiocareMedical, Lake Concord, CA, USA) and incubated for 30 min in a humid chamber. The slides were then incubated with polyclonal rabbit antibodies anti-IFNγ and TNFα (Santa Cruz Biotechnology, Santa Cruz, CA, USA) for four h at room temperature. Subsequently, slides were washed and incubated with goat anti-rabbit antibodies labelled with horseradish peroxidase for 1 h at room temperature. After extensive washings, horseradish peroxidase was revealed with 100 μL of diaminobenzidine/H_2_O_2_ (0.4004 g diaminobenzidine + 10 mL HBS + 4 mL H_2_O_2_). Slides were washed and contrasted with hematoxylin. IFNγ and TNFα positive cells in five random areas at 40× magnification for the lung were counted using Image J [51].

### 4.13. Real-Time PCR Expression Analysis of Cytokines in Infected Lungs

According to the manufacturer’s instructions, total mRNA from paraffin-embedded tissues was obtained with The NucleoSpin totalRNA FFPE kit for RNA extraction (Macherey-Nagel Thermo Fisher Scientific, Waltham, MA, USA). The quality and quantity of RNA were evaluated through spectrophotometry (260/280) and on agarose gels. Reverse mRNA transcription was performed using 2 ng RNA, oligo (dT), and the Omniscript kit (Qiagen, Hilden, Germany). RT-PCR was carried out using the 7500 real-time PCR system (Applied Biosystems Inc., Foster City, CA, USA) and Quantitect SYBR Green Master Mix kit (Qiagen, Hilden, Germany). Specific primers were designed using the program Primer 359 Express (Applied Biosystems, Waltham, MA, USA), for the following targets: RPLP0, housekeeping gene (ribosomal protein lateral stalk subunit P0), TNFα F: 5-CTCTCGCTTTCTGGAGGGTG-3; R: 5-357 ACGCGCTTGTACCCATTGAT-3, IFN-γ F: 5-GGTGACATGAAAATCCTGCAG-3; R: 5-CCTCAAACTTGGCAATACTCATGA-3. Initial denaturation at 95 °C for 15 min was followed by 40 cycles at 95 °C for 20 s, 60 °C for 20 s, and 72 °C for 34 s. Each sample was examined twice. The 2^−(∆∆Ct)^ technique calculates the fold change in gene expression [52].

### 4.14. Structural Modeling on a Naturally Occurring Water Buffalo Peptide, WBCATH

Structural modeling was undertaken on a naturally occurring peptide, WB14, comprised of the C-terminal 14 amino acid residues (positions 131–144, sequence: GLPWILLRWLFFRG) from *Bubalus bubalis* cathelicidin 4 (GenBank registry ID: AIZ93880.1) to assess its prospective mechanism of broad-spectrum antimicrobial activity. The combination of substantial hydrophobicity and an amino acid sequence commensurate with forming an amphiphilic helix (Figure 9A) led us to apply gaussian accelerated molecular dynamics (GAMD) simulations [53].

To investigate whether antimicrobial activity might arise from the assembly of peptide oligomeric associations on, or spanning a representative bilayer bacterial cell membrane. For simulation purposes, the membrane model was constructed as a 3:1 ratio between POPE and POPG, with water solvation, fully neutralizing Na+ and Cl^−^ counter-ions and a 20 Å vertical buffer from the next unit cell. All structural models were assembled using the CHARMM GUI [54]. This resulting model was then used to perform a 5 × 10^7^ step GAMD calculation to simulate the approach and interaction of an anti-parallel wb14 dimer with the membrane. Further simulations (2 × 10^8^ GAMD steps) were run to assess the stability of hypothetical hexameric, octameric and decameric transmembrane spanning WBCATH pores.

### 4.15. Statistical Analysis

In the in vitro experiments, the mean and standard error of the mean (SEM) from four separate experiments with three replicates represents the data. In the in vivo experiments, the mean and SEM from 6 individual mice in two separate experiments represents the data. All data collecting was performed in random order. The Shapiro–Wilk normality test was used to determine the normality of the data. The statistical significance of the experiments was determined using Unpaired *t*-tests (comparison of each group to the respective control). The production of TNFα and IL-12 by macrophages was analyzed by a one-way ANOVA and Tukey’s multiple comparisons test. For all experiments, statistical significance was established at *p* < 0.05. GraphPad Prism was used to conduct the statistical analysis. (v 9.1.1.225) (GraphPad, San Diego, CA, USA).

## 5. Conclusions

Current assessment of new therapeutic strategies against TB has found immunotherapy as a viable and potentially effective alternative. AMPs have a crucial role in the infection’s initial control due to their antimicrobial and immunoregulatory properties. WBCATH demonstrated effectiveness in reducing bacterial load and pneumonia by direct antimicrobial activity and promoting the expression of protective cytokines in vitro by alveolar macrophages and in murine models of progressive disease. The synergistic effect of WBCATH with first-line antibiotics is promising when looking for better treatments that can shorten chemotherapy (Figure 11).

## Figures and Tables

**Figure 1 antibiotics-12-00075-f001:**
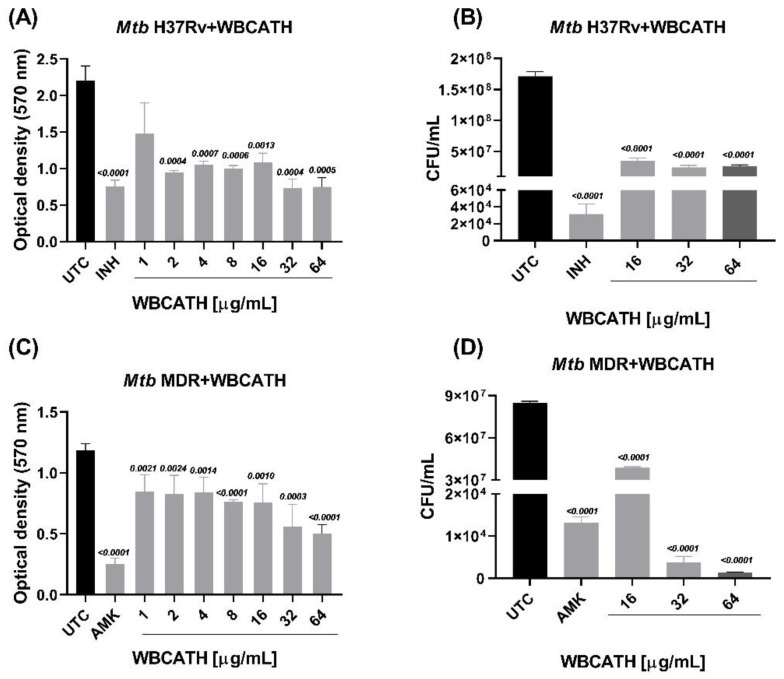
In vitro antimycobacterial activity of different concentrations of WBCATH on H37Rv and the MDR strains. (**A**) Antimycobacterial action of WBCATH on *Mtb* H37Rv tested by MIC assay. (**B**) Colony-forming units (CFU) of H37Rv *Mtb* after incubation with the indicated concentrations of WBCATH. (**C**) Antimycobacterial activity of WBCATH on *Mtb* MDR tested by MIC assay. (**D**) CFUs of MDR *Mtb* after incubation with WBCATH. There is a dose-dependent effect of WBCATH against the drug-susceptible H37Rv strain and the MDR strain in the MIC assay that was confirmed by the CFU count, being the treatment lesser effect on the MDR strain. Positive controls: isoniazid (INH) and amikacin (AMK). *t* test against untreated control UTC (*n* = 6).

**Figure 2 antibiotics-12-00075-f002:**
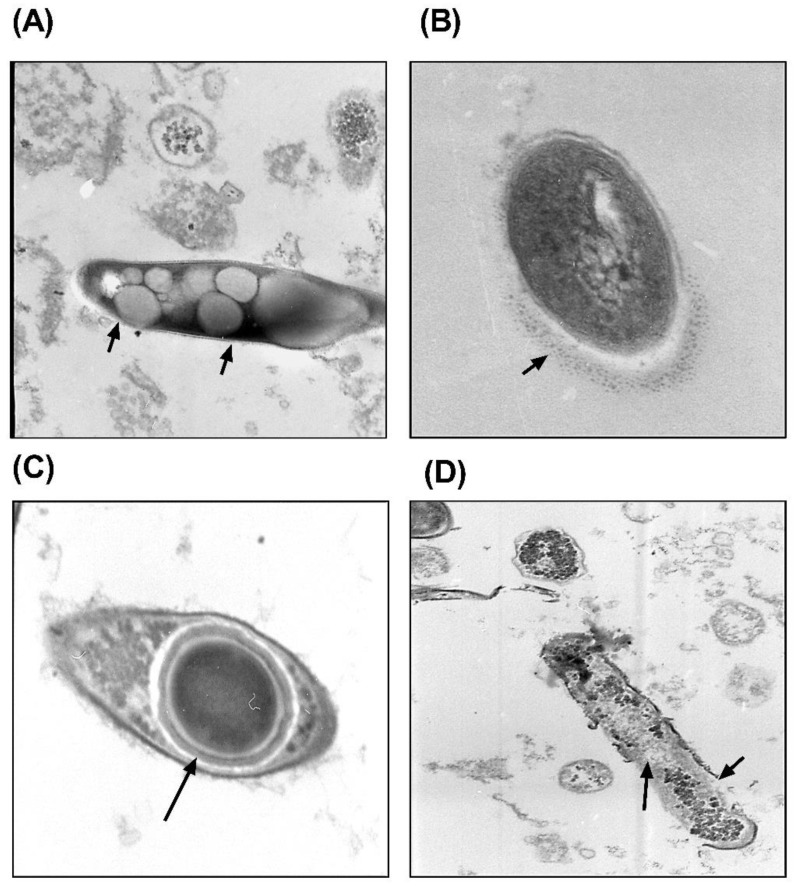
Representative electron microscopy micrographs of *Mtb* H37Rv treated with 32 and 64 μg of WBCATH. (**A**) Numerous cytoplasmic vacuoles. (**B**) Swollen and fragmentation of the bacterial cell wall (arrow) (**C**) Spherical electron-dense body surrounded by a ring of electron-dense material (arrow) and a clear halo with irregular electron-dense material dispersedly distributed in the cytoplasm. (**D**) Disruption and fragmentation of the cell wall and peripheral membrane (arrows).

**Figure 3 antibiotics-12-00075-f003:**
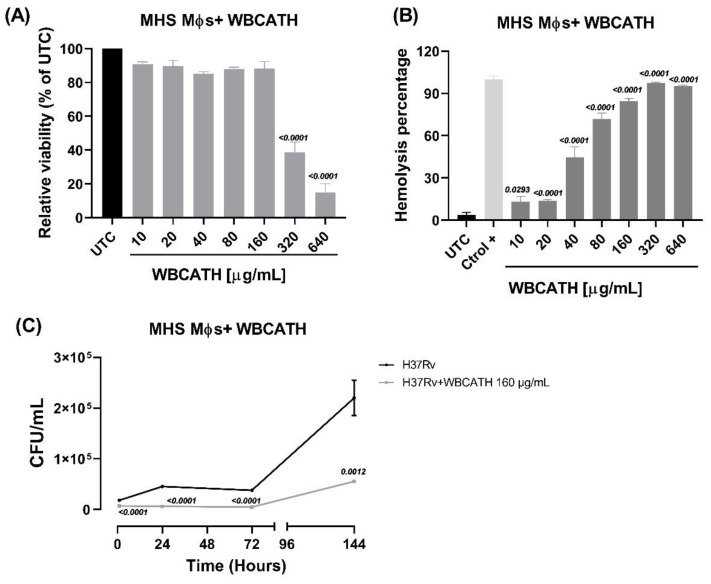
Effect of WBCATH on cell viability of non-infected and infected alveolar macrophages and erythrocytes. (**A**) WBCATH does not have cytotoxic activity against non-infected macrophages at low concentrations. *t* test against untreated control (UTC) (*n* = 6). (**B**) Hemolytic assay at different concentrations of WBCATH on freshly collected human red blood cells (**C**) Effect of WBCATH (160 µg/mL) on the bacterial burden in macrophages infected with H37Rv *Mtb*. WBCATH did not affect the survival of alveolar macrophages and induced hemolysis at high concentrations. WBCATH induced a significantly lower bacillary load from 1 to 72 h post-infection than the untreated group. *t* test against the control group (*n* = 6).

**Figure 4 antibiotics-12-00075-f004:**
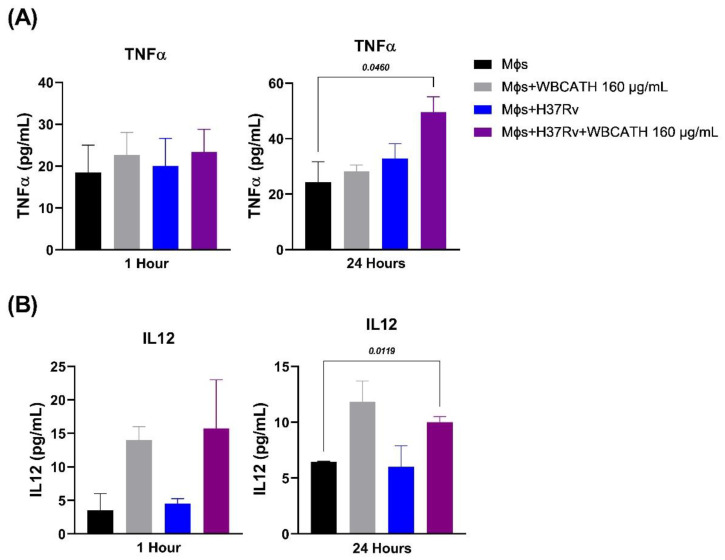
Effect of WBCATH (160 µg/mL) on the production of cytokines in non-infected and infected alveolar macrophages with *Mtb* H37Rv strain. (**A**) TNFα production after 1 and 24 h of treatment with WBCATH. (**B**) IL-12 production after 1 and 24 h of treatment with WBCATH The macrophages were incubated with WBCATH during the indicated times and in the supernatant was determined by ELISA the production of TNFα and IL-12. The treatment with WBCATH increased the production of both proinflammatory cytokines in no infected and infected macrophages. One-way Analysis of variance (ANOVA) and Tukey’s multiple comparisons test (*n* = 6).

**Figure 5 antibiotics-12-00075-f005:**
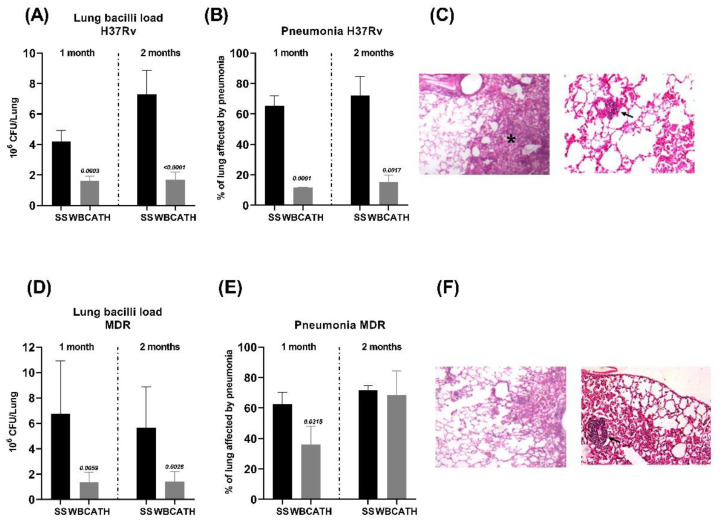
Effect of WBCATH in a murine model of pulmonary TB infected with a drug-sensible or drug-resistant strain. (**A**) Lung bacilli loads from TB mice infected with H37Rv and treated after 60 days of infection with WBCATH during 1 and 2 months (grey bars), in comparison with control animals that received only the vehicle saline solution (black bars). (**B**) Percentage of lung surface affected by pneumonia in TB mice infected with H37Rv and treated with WBCATH. Each bar represents the mean and standard deviation of six mice, *t* test. (**C**) Representative micrographs of the lung (H/E staining) left panel shows extensive areas of pneumonia (asterisk) in the control mice after four months of infection with H37Rv strain, while the right panel shows the lung of treated animals with WBCATH for two months that exhibit lesser pulmonary consolidation and blood vessel surrounded by lymphocytes (arrow). (**D**) Lung bacilli loads of mice infected with MDR strain and treated with WBCATH for 1 and 2 months (grey bars) in comparison with control non-treated animals (black bars). (**E**) The percentage of lung surface affected by pneumonia comparing WBCATH treated, and control mice (SS) affected by pneumonia. Each bar represents the mean and standard deviation of six mice, *t* test. (**F**) Representative micrographs (H/E staining) of the lung from tuberculous mice treated (right panel) or not (left panel) with WBCATH, treated mice showed similar or more pneumonia and nodules of lymphocytes around blood vessels (arrow).

**Figure 6 antibiotics-12-00075-f006:**
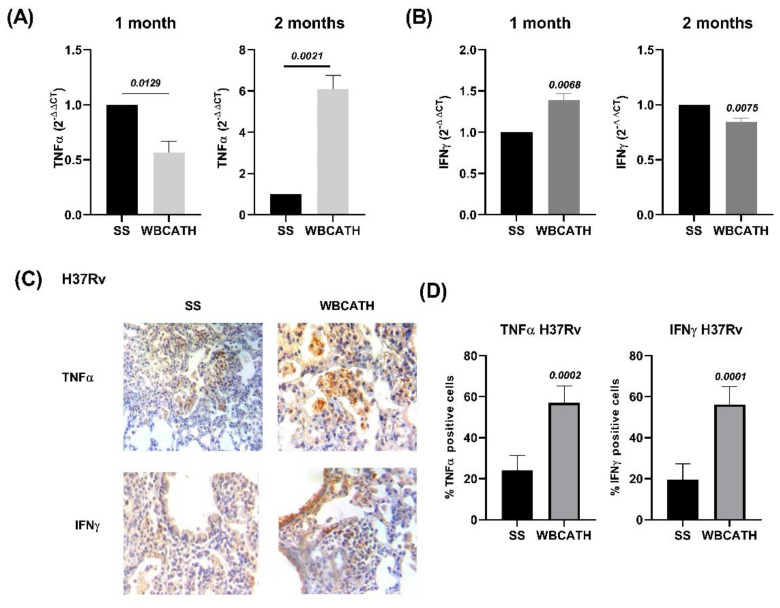
Production of pro-inflammatory cytokines in the lungs of mice infected with drug sensible strain H37Rv and treated with WBCATH. (**A**) Lung gene expression of TNFα determined by RT-PCR in tuberculous mice after 1 and 2 months of treatment with WBCATH (grey bars) in comparison with control animals that only received the vehicle saline solution (black bars). (**B**) Lung gene expression of IFNγ after 1 and 2 months of therapy with WBCATH. (**C**) Representative micrographs of immunohistochemistry to detect TNFα and IFNγ in the lung of tuberculous mice treated or not with WBCATH, treated mice showed more IFNγ immunostained cells in lymphocytes around blood vessels and bronchial epithelium, while in the pneumonic areas there were more TNFα immunostained macrophages. (**D**) These observations were confirmed by the cell count and percentage of positive cells that showed significantly higher percentages in the treated mice (*n* = 6). Bars represent the mean and standard deviation, and at the top of the bars are noted the significance of the *T* student test.

**Figure 7 antibiotics-12-00075-f007:**
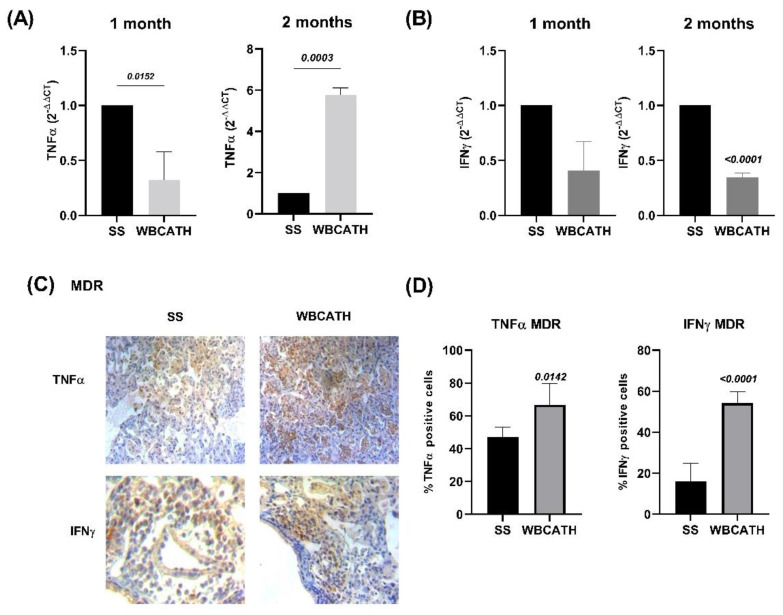
Production of protective cytokines in the lungs of MDR tuberculous mice treated with WBCATH. (**A**) Gene expression of TNFα after 1 and 2 months of treatment with WBCATH (grey bars), in comparison with control non-treated animals (black bars) (**B**) Lung gene expression of IFNγ in non-treated (black bars) and after 1 and 2 months of therapy with WBCATH (grey bars). (**C**) Representative micrographs of the cellular cytokine detection by immunohistochemistry, there are more TNFα immunostained macrophages in the pneumonic patches and IFN-γ immunoreactive lymphocytes around blood vessels in the treated mice. (**D**) The percentage of immunostained cells to TNFα and IFNγ in control and treated mice. Bars represent the mean and standard deviation of six animals. The value of statistical significance obtained by the student *T*-test is at the top of the bars.

**Figure 8 antibiotics-12-00075-f008:**
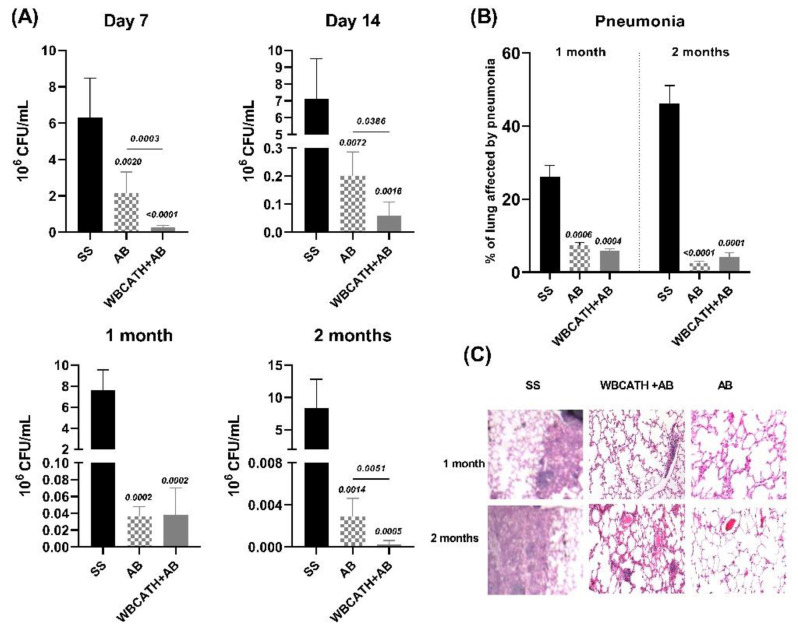
The therapeutic effect of WBCATH administration combined with first-line antibiotics (AB) on mice infected with H37Rv. (**A**) Comparison of lung bacilli loads of TB mice infected with H37Rv that received only the vehicle as the control group (SS), with the groups treated with conventional chemotherapy (AB) and WBCATH plus first-line antibiotics (WBCATH + AB), after the indicated time points. The combined treatment produced a significant reduction of bacillary loads at 7, 14 and 60 days. (**B**) The percentage of lung surface affected by pneumonia determined by automated morphometry showed a significant reduction in both treated groups in comparison with the control, but no difference was seen between the AB and WBCHAT groups. Bars represent the means and standard deviation of 6 mice per group. Statistical significance is noted in the top of the bars, Student *T* test. (**C**) Representative micrographs of the lung comparing the three groups.

**Figure 9 antibiotics-12-00075-f009:**
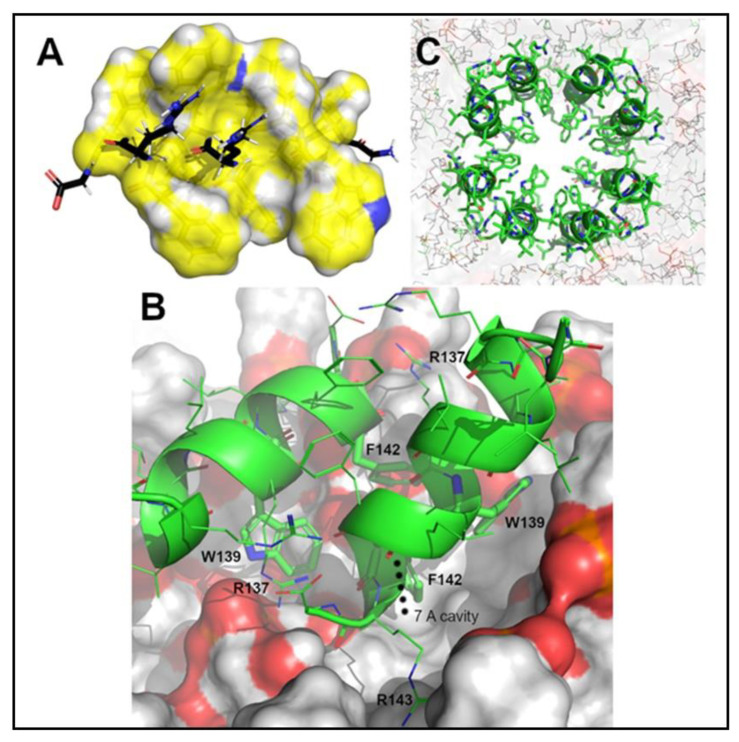
Antimicrobial peptide C-terminal fragment from *Bubalus bubalis* cathelicidin 4. (**A**) Structure A indicates that, although largely hydrophobic (yellow surfaces), a helical conformation aligns polar groups (black sticks) along a straight band to produce an amphiphilic moment commensurate with oligomeric membrane association. (**B**) The peptide associates with the surface and could excavate a cavity with a depth of roughly 7 Å below the median plane of undulation of the bilayer surface. (**C**) The octameric peptide structure sustained an ordered arrangement, reminiscent of a physiological pore.

**Figure 10 antibiotics-12-00075-f010:**
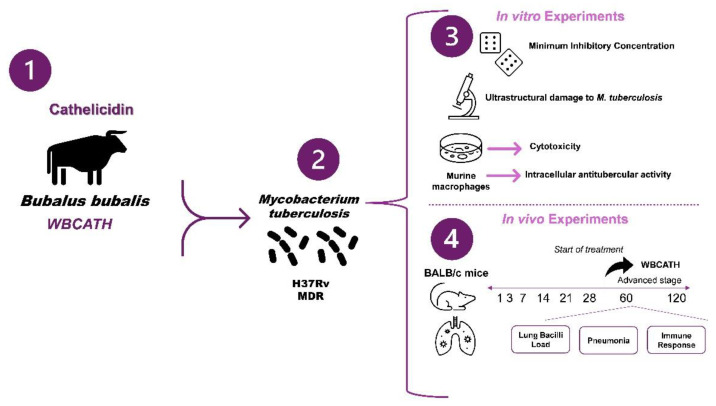
The experimental design was used to evaluate the effect of mammals cathelicidins against *Mtb.* (**1**) The WBCATH cathelicidin (**2**) The effect of this cathelicidin was evaluated against *Mtb* H37Rv, which is drug-susceptible, and the MDR clinical isolate CIBIN/UMF: 15:99. (**3**) In the first part of the research, cathelicidins were tested in vitro. These experiments include the determination of the MIC and the effect on the ultrastructure of *Mtb* of WBCATH. Then, we evaluated the effect on murine macrophages infected with *Mtb.* (**4**) In the work’s second part, we evaluated the effect of WBCATH in a murine model of pulmonary TB in terms of lung bacilli load, pneumonia, and the immune response. All experiments were performed in duplicate.

**Figure 11 antibiotics-12-00075-f011:**
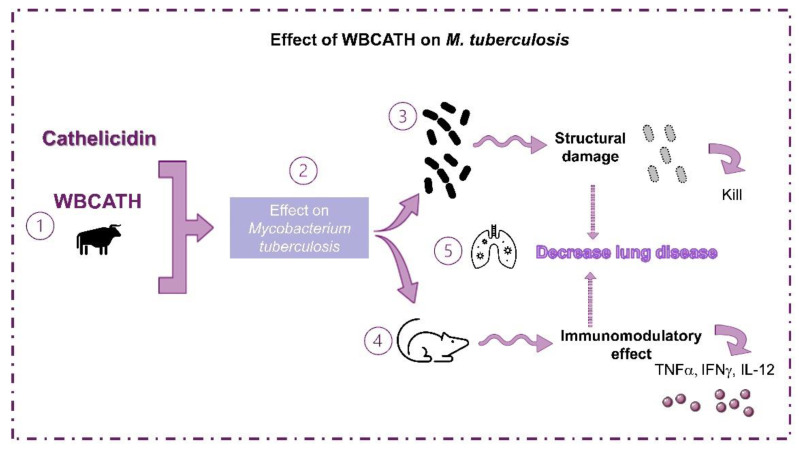
Effect of antimicrobial peptide WBCATH on *Mtb.* (**1**) In the present study, we evaluated the effect of WBCATH on *Mtb*. (**2**) Two strains of *Mtb* were used, the drug sensible mycobacteria H37Rv and MDR mycobacteria. (**3**) WBCATH induced direct damage in the mycobacteria, induced the direct killing of the bacteria, and increased the macrophage’s clearance of intracellular mycobacteria. (**4**) Furthermore, when we evaluated the treatment with WBCATH in a murine model of advanced TB, the cathelicidin used had an effective activity in reducing the bacterial load and pneumonia by direct antimicrobial action and by promoting the expression of TNFα, IFNγ and IL-12. The treatments presented a synergistic effect with first-line anti-TB drugs. (**5**) All these results show that WBCATH is an effective treatment for lung disease in advanced TB in the murine model.

**Table 1 antibiotics-12-00075-t001:** Characteristics of the *M. tuberculosis* strains used in the present work. The sensitivity or resistance to antibiotics is shown and the MIC of each antibiotic. Modified of Molina-Salinas et al., 2006 [30].

Drug	*M. tuberculosis* H37Rv	*M. tuberculosis* CIBIN/UMF 15:99 (MDR)
Streptomycin	Sensitive (0.5 μg/mL)	Resistant (>100 μg/mL)
Isoniazide	Sensitive (0.06 μg/mL)	Resistant (3.13 μg/mL)
Rifampin	Sensitive (0.06 μg/mL)	Resistant (100 μg/mL)
Ethambutol	Sensitive (2 μg/mL)	Resistant (8 μg/mL)

## Data Availability

The data is accessible on request to the corresponding authors.

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
