# Peer review of "In Vitro, In Vivo and In Silico Assessment of the Antimicrobial and Immunomodulatory Effects of a Water Buffalo Cathelicidin (WBCATH) in Experimental Pulmonary Tuberculosis"

_antibiotics, 2022, doi:10.3390/antibiotics12010075_

Round 1

Reviewer 1 Report

Palacios et al. studied the potential of water buffalo cathelicidin (WBCATH) as antimicorbial agent against pulmonary tuberculosis. The study shows promising results of WBCATH against tuberculosis either alone or combination with antibiotics. Several issues should be addressed before considering the manuscript for publications:

1)     All the figures are of low resolution which make it hard to read.

2)     The author claims a significant in vitro microbicidal effect of WBCATH, However Figure 1B as an example shows around less than 1 log CFU/mL reduction which is not efficient and indicate a lot of remaining bacteria. The exact CFU reduction values and how it compares to controls should also be discussed in the manuscript.

3)     Hemolytic activity of WBCATH should be studied

4)     Colony counting data for control antibiotics should be provided in Figure 1.

5)     The exact reduction of intracellular microbial burden (Figure 3B) should be discussed with numbers and percentages in the manuscript and compared to the controls.

6)     In Vitro inflammatory response of either infected or uninfected macrophages treated WBCATH should be provided.

Author Response

Manuscript ID: antibiotics-2091673

Title: In vitro, in vivo and in silico assessment of the antimicrobial and  immunomodulatory effects of a water buffalo cathelicidin (WBCATH) in  experimental pulmonary tuberculosis.

Reply to Reviewer #1

  • All the figures are of low resolution which make it hard to read.

Answer:  Thank you for your advice.  We have already improved the resolution of the images; we also attached the image files in high resolutions. Previously, it was a mistake in the pdf file.

  • The author claims a significant in vitro microbicidal effect of WBCATH, However Figure 1B as an example shows around less than 1 log CFU/mL reduction which is not efficient and indicate a lot of remaining bacteria. The exact CFU reduction values and how it compares to controls should also be discussed in the manuscript.

Answer: The data of CFU was represented as 106 CFU/mL, not in Log. Now it’s represented as CFU/mL. We added the information request about the reduction in lines 117-127 and is clearer the reduction of the CFU in contrast to the controls than before.  

  • Hemolytic activity of WBCATH should be studied.

 Answer: We performed hemolysis assays to observe the effect of WBCATH on human erythrocytes, the results are shown in graph 3B and the method is described in the corresponding Material and Methods section. The description of the results is in lines 159-164.

  • Colony counting data for control antibiotics should be provided in Figure 1.

Answer: We have added the requested information in Fig 1B and 1D, and the CFU of INH and AMK (positive controls) were added to the graph.

  • The exact reduction of intracellular microbial burden (Figure 3B) should be discussed with numbers and percentages in the manuscript and compared to the controls.

Answer: We added information about numbers and percentages for comparison of the different groups. Lines 167-172.

  • In Vitro inflammatory response of either infected or uninfected macrophages treated WBCATH should be provided.

Answer: We repeated the intracellular bacterial killing assay with infected and uninfected macrophages treated with WBCATH, the supernatants were collected and used to determine the cytokines TNFα and IL-12, the results are shown in Fig 4. The description of these results is in lines 182-193.

We thank reviewers for their time in dealing with our work. Thank you very much for your help. We hope that our responses will satisfy reviewers concerns. As a result, we think that a much-improved manuscript has been generated.

Reviewer 2 Report

The topic is very interesting and all of improvements my be considered. The article is well written and final results seem promising.

Some theoretical approaches seems unclear. For example: it is an experimental approach for demonstrating the innate immunity effect of cathelicidin on MT infected mice, however all experimental measurements are taken in animal models showing advanced TB disease (60 days population surviving after infection). The effects shown doesn´t refer to the first steps of TB infection. Only two TB strains are applied to achieve the final conclusions.

Authors must explain the expected hystopathologic effects of TB infection in the animal model choosen for these experiments: allways pneumonia or other focal lesions in the lung.

A more detailed WBCATH protein characterization (2D PAGE, monomer number determination) is desirable.

REVIEW

ABSTRACT

Abstract doesn´t reflect  the steps and methods expent to achieve final results.

Lines 20-21: What is the meaning of this paragraph?

INTRODUCTION

M&M

I don´t know the real reason to place M&M section between the  Discussion and Conclusions. This section must be placed before the results section. For an experimental work this condition is very important.

Lines 387-389: authors must explain why they are chosen this two MT strains and the sensibility features of them against antibiotics. MDR strain features are not detailed. Why only two strains?

Line 414: “Isoniazid or amikacin was a positive control”: this expression must be explained.

Line 419: “MTS” is not explained.

Lines 423-424 and also 493: perhaps the colony count obtained in MDR strains may be increased if agar plates are incubated more than 21 days. MDR strains displays slower metabolic pathways and a lower growing rate.

Lines 474-475: if cathelicidin prevent MT infection or enhances innate inmunity ¿what are the reasons to select the 60 days population surviving after infection? See lines 58-60 of the Introduction section.

Lines 477 and 483: what is the meaning of “vehicle”?

Lines 495-498: What is the staining to detect TB bacilli applied? Hematoxilin-Eosin stain?

RESULTS

Lines 105-106: a more detailed information about MDR isolate (only one) is lacking. Other MT wild isolates are needing to assess cathelicidin activity, because clinical isolates are sensitive in a majority of patients.

Figure 1 (B): the order of WBCATH values on the axis is different to the previous(A). Why? The colony count  data about the controls INH and AMK are not shown in  B and D. Some explanation must be done.

Figure 2 and lines 121-128: Which type of cells show the abnormalities? Are there these kind  of abnormalities in control isoniazid and amikacin?

Figure 3: in (B); what is the curve of the bacillary load without WCATH in infected macrophages?

Figure 5: (C) experiment displaying is unclear. Perhaps another approach, for example by detecting INF and TNF proteins by SDS-PAGE method and quantifying the corresponding protein bands. The experiment shows mostly  the TNF or INF location.

Figures 5 (B) and 6 (B): there are important differences, therefore some explanation might be done.

Lines 242-267: the description of the computational model of WCATH must be completed  showing more data about protein characterization: molecular mass, SDS-PAGE analysis of monomers, 2D PAGE molecular oligomer: ¿why 8 monomers?.

DISCUSSION

Lines 330-331: according to the results shown in Figures 5 (B) and 6 (B), this conclusion is not supported.

Lines 369-372: final conclusions must be lowered because:

-          There is an animal model experiment. TB infection in human can be a different concern.

-          Only two MT strains has been proven: WBCATH has a different pro-inflammatory effect in  MDR strain than in H37Rv strain.

Author Response

ABSTRACT

Abstract doesn´t reflect the steps and methods expent to achieve final results.

Lines 20-21: What is the meaning of this paragraph?

Answer: We have corrected it and added the steps and methods performed to achieve our final results.

  • M&M

I don´t know the real reason to place M&M section between the Discussion and Conclusions. This section must be placed before the results section. For an experimental work this condition is very important.

Answer: We agree with the reviewer but the journal format establishes that M&M should be at the end of the manuscript.

  • Lines 387-389: authors must explain why they are chosen this two MT strains and the sensibility features of them against antibiotics. MDR strain features are not detailed. Why only two strains?

Answer: we select the drug-susceptible laboratory reference strain (H37Rv, ATCC 27294) because it is sensible to all the primary line antibiotics: streptomycin, isoniazid, rifampin, ethambutol and pyrazinamide, this Mtb strain has been used since the beginning of the characterization of the used murine TB model and in several studies that evaluate the activity of diverse antimicrobial-peptides ((J Leukoc Biol. 2021 Nov;110(5):951-963. doi: 10.1002/JLB.4MA0920-627R; Int J Antimicrob Agents. 2013 Feb;41(2):143-8. doi: 10.1016/j.ijantimicag.2012.09.015). The MDR strain of M. tuberculosis CIBIN/UMF: 15:99 is resistant to all the mentioned antibiotics; it is a clinical isolate from a patient from the north of Mexico and this strain has been used extensively in this kind of study. The reference to its microbiology characteristics is added and all this additional information is now mentioned in the corresponding M&M section. Unfortunately, the extra time conceded by the editor is not enough to prepare more strains and determine their MIC.

  • Line 414: “Isoniazid or amikacin was a positive control”: this expression must be explained.

Answer: We have explained this in lines 493-494

  • Line 419: “MTS” is not explained.

Answer: We have corrected and explained this sentence in lines 497-504

  • Lines 423-424 and also 493: perhaps the colony count obtained in MDR strains may be increased if agar plates are incubated more than 21 days. MDR strains displays slower metabolic pathways and a lower growing rate.

Answer: It is possible but this is the usual or conventional time of incubation and unfortunately we do not have more lung homogenates to grow bacteria for more than 21 days and repeating these experiments take too much time. On the other hand, after three weeks the culture medium loses nutrients, humidity and this affects the growth of the bacteria (J Antimicrob Chemother, 66(10), 2277–2280. https://doi.org/10.1093/jac/dkr288).

  • Lines 474-475: if cathelicidin prevent MT infection or enhances innate inmunity ¿what are the reasons to select the 60 days population surviving after infection? See lines 58-60 of the Introduction section.

Answer: In this murine TB model there is a high production of cathelicidin during the first month of infection, followed by a decrease during late active disease ((Clin Exp Immunol 2010 Sep; 161(3):542-50. doi: 10.1111/j.1365-2249.2010.04199.x). Thus, at day 60 of infection lung damage is extensive (pneumonia) and the production of cathelicidin is reduced, the administration of water buffalo cathelicidin started at this time because the aim was to evaluate its therapeutic activity when the lung production of cathelicidin is low and is ongoing the late active disease with extensive tissue damage that is similar that patients with advanced disease.

  • Lines 477 and 483: what is the meaning of “vehicle”?

Answer: Vehicle is the solution used to suspend the cathelicidin for its administration, which in this case was the saline solution, now after the vehicle in parenthesis is written saline solution line 581

  • Lines 495-498: What is the staining to detect TB bacilli applied? Hematoxilin-Eosin stain?

Answer: We did not used any staining method to detect TB bacilli. Hematoxylin and Eosin is not useful to stain mycobacteria.

  • RESULTS

Lines 105-106: a more detailed information about MDR isolate (only one) is lacking. Other MT wild isolates are needing to assess cathelicidin activity, because clinical isolates are sensitive in a majority of patients.

 Answer: more information and a new reference about the MDR clinical isolate is now added in the corresponding M&M section and we added the information in table 1 (Lines 460-466). In particular in-vivo experiments using the murine TB model are demanding and expensive, for these reasons we used only two strains. We agree that it is important and possible to test in-vitro more strains but the allowed conceded extra-time was not enough to do it.

  • Figure 1 (B): the order of WBCATH values on the axis is different to the previous(A). Why? The colony count data about the controls INH and AMK are not shown in B and D. Some explanation must be done.

Answer: We have added the requested information in the Fig 1B and 1D, the CFU were attached of INH and AMK (positive controls). We only do the UFC of the three highest concentrations and the order of the WBCATH in the axis was corrected to be the same in all the figures.

  • Figure 2 and lines 121-128: Which type of cells show the abnormalities? Are there these kind of abnormalities in control isoniazid and amikacin?

Answer: We have corrected this sentence indicating that drug sensible and MDR mycobacteria are the cells that show the described abnormalities. We did not perform isoniazid and amikacin controls for electron microscopy. Lines 139-144

  • Figure 3: in (B); what is the curve of the bacillary load without WCATH in infected macrophages?

Answer: A: Bacillary load determination in infected macrophages without WBCATH (control) corresponds to the black solid line and it is referred as H37Rv in Fig 3C.

  • Figure 5: (C) experiment displaying is unclear. Perhaps another approach, for example by detecting INF and TNF proteins by SDS-PAGE method and quantifying the corresponding protein bands. The experiment shows mostly the TNF or INF location.

Answer: The reviewer is right in the comment that immunohistochemistry is the method to determine the localization of cytokines production in particular cells, but at the present, the evaluation by modern automated morphometry equipment that estimates the intensity and number of the immunostained cells for specific molecules in determinate tissue areas is a reliable quantitative method, we also preferred this system that uses fixed non-contaminant tissue because it avoids the handling of tissue samples with high amount of pathogenic microorganisms.

  • Figures 5 (B) and 6 (B): there are important differences, therefore some explanation might be done.

Answer: We have corrected and explained this observation in the Discussion section, and commented on the dual activity of cathelicidin producing pro or anti-inflammatory effects.

  • Lines 242-267: the description of the computational model of WCATH must be completed showing more data about protein characterization: molecular mass, SDS-PAGE analysis of monomers, 2D PAGE molecular oligomer: ¿why 8 monomers?.

Answer: The manuscript now included a rigorous description of modeling protein in the Methods section and it is alluded to by Section number. The oligomeric dependencies have now been addressed qualitatively.

  • DISCUSSION

Lines 330-331: according to the results shown in Figures 5 (B) and 6 (B), this conclusion is not supported.

Answer: We have corrected and explained this question.

  • Lines 369-372: final conclusions must be lowered because:

Answer: We have modified the sentence.

  • There is an animal model experiment. TB infection in human can be a different concern.

Answer: MTB is a pathogen that preferentially infects human hosts. It is therefore difficult to find a model that exactly reproduces the disease that is produced in humans by this microorganism. Despite its limitations, the murine model has long been one of the most widely used and successful for expanding our knowledge of the immunopathology of this disease. First, humans and mice have important similarities in the main features of the innate and adaptive immune responses to mycobacteria. In this regard, we can cite the protective role of CD4+ T lymphocytes, and of IFN-γ and TNF-α. Additionally, there is a high availability of immunological reagents and methodologies adapted to the murine model, these being, due to their size and reproduction characteristics, one of the models with the lowest cost compared to other animal species. Finally, genetic standardization in various mouse strains decreases the variation between subjects, thus obtaining more precise results (Tuberculosis (Edinburgh, Scotland), 2009 89(3), 195–198. https://doi.org/10.1016/j.tube.2009.02.002).

  • Only two MT strains has been proven: WBCATH has a different pro-inflammatory effect in  MDR strain than in H37Rv strain.

   Answer: We agree with this comment, the final paragraph of the Discussion was modifiedaccording to these important points

We thank reviewers for their time in dealing with our work. Thank you very much for your help. We hope that our responses will satisfy reviewers concerns. As a result, we think that a much-improved manuscript has been generated.

Round 2

Reviewer 1 Report

The authors replied adequately for the raised issues. I can recommend this article for publication.